# Delayed antibiotic exposure induces population collapse in enterococcal communities with drug-resistant subpopulations

Kelsey M Hallinen[1†], Jason Karslake[1†], Kevin B Wood[1,2]*

[1]Department of Biophysics, University of Michigan, Ann Arbor, United States;
[2]Department of Physics, University of Michigan, Ann Arbor, United States

**Abstract** The molecular underpinnings of antibiotic resistance are increasingly understood, but less is known about how these molecular events influence microbial dynamics on the population scale. Here, we show that the dynamics of *E. faecalis* communities exposed to antibiotics can be surprisingly rich, revealing scenarios where increasing population size or delaying drug exposure can promote population collapse. Specifically, we demonstrate how density-dependent feedback loops couple population growth and antibiotic efficacy when communities include drug-resistant subpopulations, leading to a wide range of behavior, including population survival, collapse, or one of two qualitatively distinct bistable behaviors where survival is favored in either small or large populations. These dynamics reflect competing density-dependent effects of different subpopulations, with growth of drug-sensitive cells increasing but growth of drug-resistant cells decreasing effective drug inhibition. Finally, we demonstrate how populations receiving immediate drug influx may sometimes thrive, while identical populations exposed to delayed drug influx collapse.

**\*For correspondence:**
kbwood@umich.edu

[†]These authors contributed equally to this work

**Competing interests:** The authors declare that no competing interests exist.

## Introduction

Antibiotic resistance is a growing public health threat (*Davies and Davies, 2010*). Decades of rapid progress fueled by advances in microbiology, genomics, and structural biology have led to a detailed but still growing understanding of the molecular mechanisms underlying resistance (*Blair et al., 2015*). At the same time, recent studies have shown that drug resistance can be a collective phenomenon driven by emergent community-level dynamics (*Vega and Gore, 2014*; *Meredith et al., 2015b*). For example, drug degradation by a sub-population of enzyme-producing cells can lead to cooperative resistance that allows sensitive (non-producing) cells to survive at otherwise inhibitory drug concentrations (*Yurtsev et al., 2013*; *Sorg et al., 2016*; *Yurtsev et al., 2016*). Additional examples of collective resistance include density-dependent drug efficacy (*Brook, 1989*; *Udekwu et al., 2009*; *Tan et al., 2012*; *Karslake et al., 2016*), indole-mediated altruism (*Lee et al., 2010*), and increased resistance in dense surface-associated biofilms (*Davies, 2003*). The growing evidence for collective resistance underscores the need to understand not just the molecular underpinnings of resistance, but also the ways in which these molecular-level events shape population dynamics at the level of the bacterial community. Indeed, a wave of recent studies are inspiring novel strategies for combating resistance by exploiting different features of the population dynamics, ranging from competition for resources (*Hansen et al., 2017*; *Hansen et al., 2019*) or synergy with the immune system (*Gjini and Brito, 2016*) to temporal and spatial features of growth, selection, or the application of drug (*Lipsitch and Levin, 1997*; *Meredith et al., 2015a*; *Fuentes-Hernandez et al., 2015*; *Zhang et al., 2011*; *Baym et al., 2016a*; *Greulich et al., 2012*;

**eLife digest** Antibiotic resistance is a threat to human and animal health worldwide. Although we rely on antibiotics to treat diseases caused by bacteria, such as tuberculosis, some bacteria are already resistant to many of the drugs available. Understanding the basis of resistance is crucial for developing new antibiotics, and for using current drugs more efficiently.

One way that bacteria resist antibiotics is by producing enzymes that inactivate specific drugs. If a community of bacteria contains both vulnerable and resistant cells, this can lead to a phenomenon called 'cooperative resistance'. When treated with antibiotics, vulnerable cells within the group are shielded by their resistant neighbors, which effectively remove the drugs from the environment.

Cooperative resistance can make it difficult for researchers to understand how resistance develops in different bacterial populations. This is because a large group of cells may collectively behave in a different way than individual cells. This means that bacterial populations are a more realistic model for 'real-world' infections and disease than studies of single cells. Now, Hallinen, Karslake and Wood show how cooperation between cells affects the way bacterial communities respond to beta-lactams, the most commonly prescribed class of antibiotic drugs.

Experiments using cultures of *Enterococcus faecalis*, a bacterium that often causes hospital infections, revealed that the density of different bacterial populations changes the effectiveness of drugs. Although increased cell density had a protective effect on populations containing only resistant bacteria, it made non-resistant populations even more vulnerable.

Mathematical modelling using information from the culture experiments predicted that interactions between vulnerable and resistant bacteria within a mixed community can determine how populations change over time. For example, if the number of antibiotic-sensitive cells is too high, this can cause the entire population to collapse. These predictions contradict the conventional understanding of how antibiotic resistance spreads, where small numbers of resistant cells multiply rapidly at the expense of vulnerable ones.

These results shed new light on the complex dynamics of antibiotic resistance within bacterial populations as a whole. In the future, they may inspire new ecology-based strategies for slowing the spread of resistance, ultimately helping reduce the burden of disease.

*Hermsen et al., 2012*; *Moreno-Gamez et al., 2015*; *De Jong and Wood, 2018*; *Trindade et al., 2009*; *Borrell et al., 2013*; *Bonhoeffer et al., 1997*; *Bergstrom et al., 2004*; *Bonhoeffer et al., 1997*; *Yoshida et al., 2017*; *Nichol et al., 2015*; *Roemhild et al., 2018*; *Baym et al., 2016b*; *Michel et al., 2008*; *Hegreness et al., 2008*; *Pena-Miller et al., 2013*; *Rodriguez de Evgrafov et al., 2015*; *Munck et al., 2014*; *Torella et al., 2010*; *Imamovic and Sommer, 2013*; *Kim et al., 2014*; *Pál et al., 2015*; *Barbosa et al., 2017*; *Barbosa et al., 2018*; *Nichol et al., 2019*; *Maltas and Wood, 2019*; *Podnecky et al., 2018*; *Imamovic et al., 2018*; *Dean et al., 2020*). As a whole, these studies demonstrate the important role of community-level dynamics for understanding and predicting how bacteria respond and adapt to antibiotics. Despite the relatively mature understanding of resistance at the molecular level, however, the population dynamics of microbial communities in the presence of antibiotics are often poorly understood.

Here we investigate dynamics of *E. faecalis* populations exposed to (potentially time-dependent) influx of ampicillin, a commonly-used β-lactam. *E. faecalis* is an opportunistic pathogen that contributes to a number of clinical infections, including infective endocarditis, urinary tract infections, and blood stream infections (*Clewell et al., 2014*; *Huycke et al., 1998*; *Hancock and Gilmore, 2006*; *Ch'ng et al., 2019*). β-lactams are among the most commonly used antibiotics for treating *E. faecalis* infections, though resistance is a growing problem (*Miller et al., 2014*). Resistance to ampicillin can arise in multiple ways, including by mutations to the targeted penicillin binding proteins or production of β-lactamase, an enzyme that hydrolyzes the β-lactam ring and renders the drug ineffective. Enzymatic drug degradation is a common mechanism of antibiotic resistance across species and has been recently linked to cooperative resistance in *E. coli* (*Yurtsev et al., 2013*) and *S. pneumoniae* (*Sorg et al., 2016*). In addition, *E. faecalis* populations exhibit density-dependent growth when exposed to a wide rang-lactamse of antibiotics (*Karslake et al., 2016*). Increasing population density typically leads to decreased growth inhibition by antibiotics, consistent with the classical inoculum

effect (IE) (*Brook, 1989*). However, β-lactams can also exhibit a surprising 'reverse' inoculum effect (rIE) characterized by increased growth of the population at lower densities (*Karslake et al., 2016*; *Jokipii et al., 1985*). In *E. faecalis*, the rIE arises from a decrease in local pH at increasing cell densities (*Karslake et al., 2016*), which are associated with increased activity of ampicillin and related drugs (*Yang et al., 2014*). Similar growth-driven changes in pH have been recently shown to modulate intercellular interactions (*Ratzke and Gore, 2018*), promote ecological suicide in some species (*Ratzke et al., 2018*), and even to modulate antibiotic tolerance in multispecies communities (*Aranda-Díaz et al., 2020*). In addition to these in vitro studies, recent work shows that *E. faecalis* infections started from high- and low-dose inocula lead to different levels of immune response and colonization in a mouse model (*Chong et al., 2017*).

In this work, we show that density-dependent feedback loops couple population growth and drug efficacy in *E. faecalis* communities comprised of drug-resistant and drug-sensitive cells exposed to time-dependent concentrations of antibiotic. By combining experiments in computer-controlled bioreactors with simple mathematical models, we demonstrate that coupling between cell density and drug efficacy can lead to rich dynamics, including bistabilities where low-density populations survive while high-density populations collapse. In addition, we experimentally show that there are certain scenarios where populations receiving immediate drug influx may eventually thrive, while identical populations exposed to delayed drug influx–which also experience lower average drug concentrations–are vulnerable to population collapse. These results illustrate that the spread of drug resistant determinants exhibits rich and counterintuitive dynamics, even in a simplified single-species population.

## Results

### Resistant and sensitive populations exhibit opposing density-dependent effects on antibiotic inhibition

To investigate the dynamics of *E. faecalis* populations exposed to β-lactams, we first engineered drug resistant *E. faecalis* strains that contain a multicopy plasmid that constitutively expresses β-lactamase (Materials and methods). Sensitive cells harbored a similar plasmid that lacks the β-lactamase insert. To characterize the drug sensitive and drug resistant strains, we estimated the half maximal inhibitory concentration, $IC_{50}$, of ampicillin in liquid cultures starting from a range of inoculum densities (*Figure 1A*; Materials and methods). We found that the $IC_{50}$ for sensitive strains is relatively insensitive to inoculum density over this range, while β-lactam producing resistant cells exhibit strong inoculum effects (IE) and show no inhibition for inoculum densities greater than $10^{-5}$ (OD units) even at the highest drug concentrations (10 μg/mL). To directly investigate growth dynamics at larger densities–similar to what can be resolved with standard optical density measurements–we used computer controlled bioreactors to measure per capita growth rates of populations held at constant densities and exposed to a fixed concentration of drug (as in *Karslake et al., 2016*). At these higher densities, we found that resistant strains are insensitive to even very large drug concentrations (in excess of $10^3$ μg/mL). By contrast, sensitive populations are inhibited by concentrations smaller than 1 μg/mL, and the inhibition depends strongly on density, with higher density populations showing significantly decreased growth (*Figure 1B*)–indicative of a reverse inoculum effect (rIE). Taken together, these results illustrate opposing effects of cell density on drug efficacy in sensitive and resistant populations. In what follows, we focus on dynamics in the regime OD > 0.05, where the interplay between these two opposing effects may dictate survival or extinction of resistant populations.

### Resistant populations exhibit bistability between survival and extinction in the presence of constant drug influx

Bacteria in natural or clinical environments may often be exposed to drug concentrations that change over time. To introduce non-constant antibiotic concentrations, we grew populations in computer controlled bioreactors capable of precise control of inflow (e.g. drug and media) and outflow in each growth chamber (*Figure 1C*; see also *Toprak et al., 2012*; *Toprak et al., 2013*; *Karslake et al., 2016*). Cell density is monitored with light scattering (OD), and each chamber received fresh media and drug at a rate $\mu_0 \approx 0.1$ hr$^{-1}$, which is approximately an order of magnitude

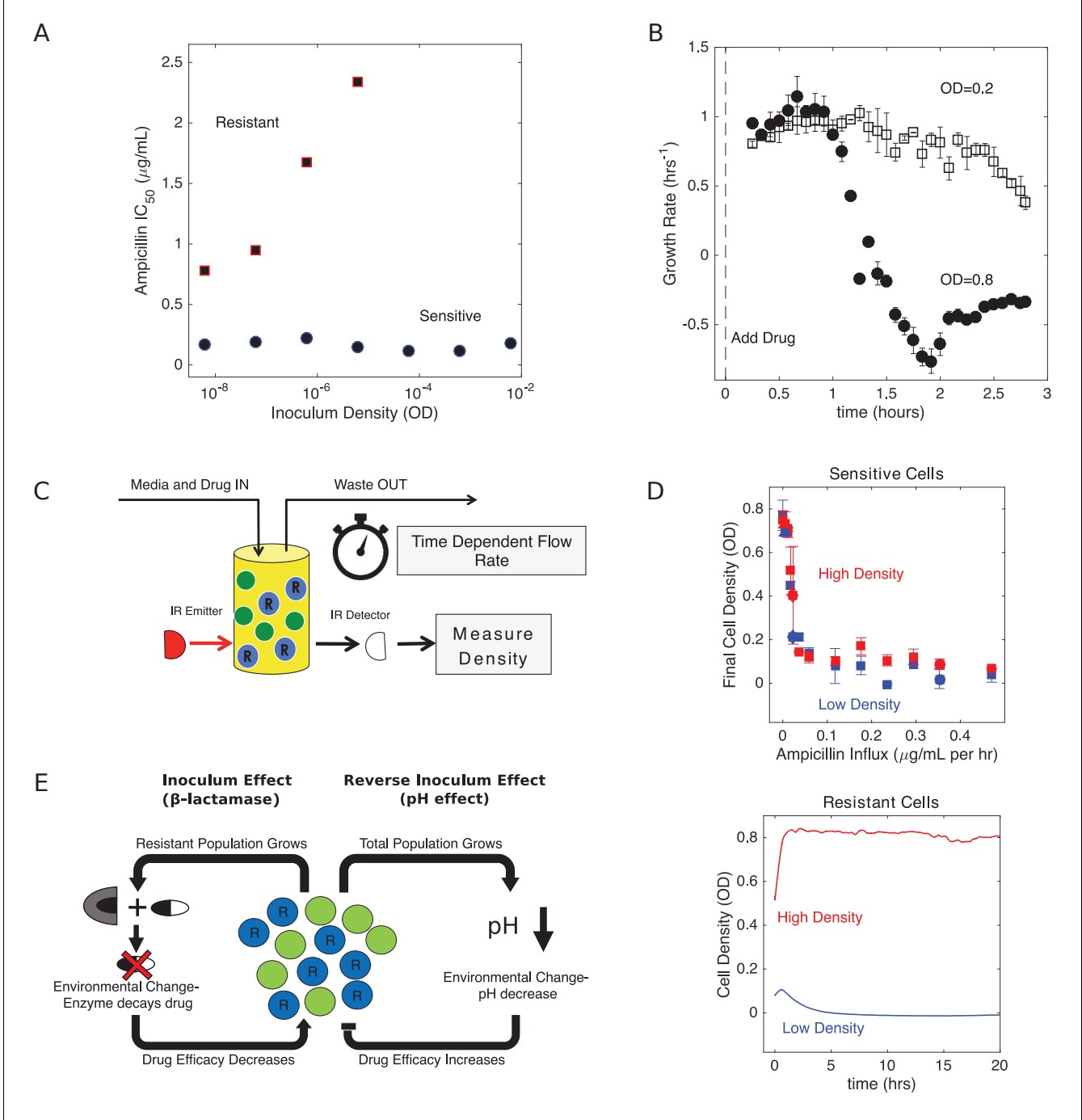

**Figure 1.** Changes in cell density have opposing effects on β-lactam efficacy in drug sensitive and drug resistant populations. (**A**) Half-maximal inhibitory concentration (IC$_{50}$) of ampicillin as a function of inoculum density for resistant (red squares) and sensitive (blue circles) populations. IC$_{50}$ is estimated using a fit to Hill-like function $f(x) = (1 + (x/K)^h)^{-1}$, where $h$ is a Hill coefficient and $K$ is the IC$_{50}$. (**B**) Per capita growth rate of drug-sensitive populations held at a density of OD = 0.2 (open squares) and OD = 0.8 (filled circles) following addition of ampicillin at time 0. Growth rate is estimated, as in *Karslake et al. (2016)*, from the average media flow rate required to maintain populations at the specified density in the presence of a constant drug concentration of 0.5 µg/mL. Flow rate is averaged over sliding 20 min windows after drug is added. Note that drug-resistant populations exhibit no growth inhibition over these density ranges, even for drug concentrations in excess of 10$^3$ µg/mL. (**C**) Schematic of experimental setup. Cell density in planktonic populations is measured via light scattering from IR detector/emitter pairs calibrated to optical density (OD). Fresh media (containing appropriate drug concentrations) is introduced over time using computer-controlled peristaltic pumps, and waste is simultaneously removed to maintain constant volume (see Materials and methods). (**D**) Top panel: final cell density of drug sensitive populations exposed to constant drug influx over a 20 hr period. Experiments were started from either 'high density' (OD = 0.6, red) or 'low density' (OD = 0.1, blue) initial populations.

*Figure 1 continued on next page*

*Figure 1 continued*

Bottom panel: cell density time series for drug-resistant populations exposed to ampicillin influx of approximately 1200 µg/mL per hour. In all experiments media was refreshed and waste removed at a rate of $\mu_0 \approx 0.1$ hr$^{-1}$. E. In mixed populations containing both sensitive (green) and resistant (blue, 'R') cells, there are opposing density-dependent effects on drug efficacy. Increasing the density of resistant cells is expected to decrease drug efficacy as a result of increased β-lactamase production (left side). By contrast, increasing the density of the total cell population decreases the local pH and increases the efficacy of β-lactam antibiotics (right side).

The online version of this article includes the following source data for figure 1:

**Source data 1.** Experimental data in *Figure 1*.

slower than the per capita growth rate of sensitive cells in drug-free media. In the absence of drug, cells reach a steady state population size of $C(1 - \mu)$, where $C$ is the carrying capacity ($C \approx 1$ in our experiments), $\mu = \mu_0/g_{max}$, and $g_{max}$ is the drug-free (maximum) per capita growth rate of bacteria. By changing the concentration of drug $D_r$ in the media reservoir, we can expose cells to effective rates of drug influx $F = \mu_0 D_r$.

We first characterized the population dynamics of each cell type (resistant, sensitive) alone in response to different influx rates of ampicillin. In each experiment, we started one population at OD = 0.6 ('high-density') and one at OD = 0.1 ('low density'). Not surprisingly, sensitive only populations exhibit a monotonic decrease in final (20 hr) population size with increasing drug concentration (*Figure 1D*, top panel), with both high- and low-density populations approaching extinction for F >0.1 µg/mL/ per hr. By contrast, high- and low-density populations of resistant cells exhibit divergent behavior, with high-density populations surviving and low-density populations collapsing (*Figure 1D*, bottom panel). In addition, we note that the resistant strains have dramatically increased minimum inhibitory concentrations (MIC), with high-density populations surviving at $D_r = 10^4$ µg/mL (an effective influx of over 1000 µg/mL/hr). Indeed, the half-maximal inhibitory concentrations (IC$_{50}$) for sensitive-only and resistant-only populations differ significantly even at very low densities (*Figure 1A*), suggesting intrinsic differences in resistance even in the absence of density-dependent coupling. This difference corresponds to a direct benefit provided to the enzyme-producing cells, above and beyond any benefit that derives from drug degradation by neighboring cells.

These results, along with those in previous studies (*Karslake et al., 2016*), are consistent with a picture of competing density-dependent feedback loops in populations comprised of both sensitive and resistant sub-populations (*Figure 1E*). Increasing the total population density potentiates the drug, a consequence of the pH-driven reverse inoculum effect (rIE). On the other hand, increasing the size of only the β-lactamase producing subpopulation is expected to decrease drug efficacy as enzymatic activity decreases the external drug concentration. These opposing effects couple the dynamics of different subpopulations with drug efficacy, which in turn modulates both the size and composition of the community.

## Mathematical model of competing density effects predicts bistability favoring survival of high-density populations at high drug influx rates and low-density populations at low influx rates

To investigate the potential impact of these competing density effects on population dynamics, we developed a simple phenomenological mathematical model that ascribes density-dependent drug efficacy to a change in the effective concentration of the antibiotic (see SI for alternative models). Specifically, the dynamics of sensitive and resistant populations are described by

$$
\begin{aligned}
\frac{dN_s}{dt} &= g(D)\left(1 - \tfrac{N_s + N_r}{C}\right)N_s - \mu N_s, \\
\frac{dN_r}{dt} &= g(D')\left(1 - \tfrac{N_s + N_r}{C}\right)N_r - \mu N_r
\end{aligned}
\tag{1}
$$

where $N_s$ is the density of sensitive cells, $N_r$ the density of resistant cells, $C$ is the carrying capacity (set to one without loss of generality), µ is a rate constant that describes the removal of cells due to (slow) renewal of media and addition of drug, $D$ is the effective concentration of drug (measured in units of MIC of the sensitive cells), and $D' = D/K_r$, where $K_r$ is a factor that describes the increase in drug minimum inhibitory concentration (MIC) for the resistant (enzyme producing) cells in low-density populations where cooperation is negligible. The function $g(x)$ is a dose response function that

describes the per capita growth rate of a population exposed to concentration $x$ of antibiotic and is given by *Udekwu et al. (2009)*:

$$g(x) = \frac{(1 - x^h)g_{max}g_{min}}{x^h g_{max} + g_{min}} \tag{2}$$

where $h$ is a Hill coefficient that describes the steepness of the dose response function, $g_{max}$ is the growth in the absence of drug, and $g_{min} > 0$ is the maximum death rate. The function $g(x)$ is a sigmoidal function that equals $g_{max}$ at $x = 0$ (no drug), decreases monotonically and crosses the horizontal axis at $x = 1$, and then approaches the maximum death rate $g_{min}$ as $x$ approaches infinity ($g(x) \rightarrow -g_{min}$). Without loss of generality, we set $g_{max} = 1$, which is equivalent to measuring all rates in time units set by $g_{max}^{-1}$ (coincidentally, we find that drug-free growth rate under the current experimental conditions is approximately $g_{max} = 1$ hr$^{-1}$, so measuring rates in units of $g_{max}^{-1}$ is equivalent to measuring time in hours).

To account for the density dependence of drug efficacy, we model the effective drug concentration as

$$\frac{dD}{dt} = F + \epsilon_1(N_s + N_r)D - \epsilon_2 N_r D - D\mu \tag{3}$$

where $\epsilon_1 > 0$ is an effective rate constant describing the reverse inoculum effect (proportional to total population size), which is modeled as an increase in the effective drug concentration with cell density. We do not mean to imply that the cells physically produce antibiotic; instead, this phenomenological model is intended to capture the increase in drug efficacy due to acidification of the local environment as density increases. Similarly, the parameter $\epsilon_2 > 0$ describes the enzyme-driven 'normal' inoculum effect (proportional to the size of the resistant subpopulation), which corresponds mathematically–and in this case, also physically–to a degradation of antibiotic. $F = D_r\mu$ is rate of drug influx into the reservoir, which can be adjusted by changing the concentration $D_r$ in the drug reservoir. When $\epsilon_2 \leq \epsilon_1$–when the per capita effect of the inoculum effect (IE) is less than or equal to that of its reverse (rIE) counterpart–the $\epsilon_1$ term is always larger in magnitude than the $\epsilon_2$ term and the net effect of increasing total cell density is to increase effective drug concentration, regardless of population composition. This regime is inconsistent with experiments, where resistant-only populations exhibit a strong IE and sensitive-only populations a rIE (*Figure 1*). We therefore focus on the case $\epsilon_2 > \epsilon_1$, where density and composition-dependent trade-offs may lead to counterintuitive behavior.

Despite the simplicity of the model, it predicts surprisingly rich dynamics (*Figure 2*). At rates of drug influx below a critical threshold ($F < F_c$), populations reach a stable fixed point at a density approaching $C(1 - \mu)$ as influx approaches zero. On the other hand, populations go extinct for large influx rates $F \gg F_c$, regardless of initial density or composition. Between the two regimes lies a region of bistability, where populations are expected to survive or die depending on the initial conditions. To characterize the behavior in this bistable region, we calculated the separatrix–the surface separating regions of phase space leading to survival from those leading to extinction–for different values of the antibiotic influx rate using an iterative bisection algorithm, similar to *Cavoretto et al. (2017)*. The analysis reveals that increasing total population size can lead to qualitatively different behavior–survival or extinction–depending on the rate of drug influx.

For influx rates at the upper end of the bistable region–and for sufficiently high initial fractions of resistant cells– high-density populations survive while low-density populations go extinct (*Figure 2*, bottom right panel). For example, in populations with an initial resistant fraction of 3/4, small populations approach the extinction fixed point while large populations are expected to survive (*Figure 2*). Intuitively, the high-density populations have a sufficiently large number of resistant cells, and therefore produce a sufficient quantity of β-lactamase, that effective drug concentrations reach a steady state value below the MIC of the resistant cells, leading to a density-dependent transition from extinction to survival as the separatrix is crossed (*Figure 2*, bottom right).

Behavior in the low-influx regime of bistability ($F \approx F_c$) is more surprising. In this regime, the model predicts a region of bistability where initially high-density populations go extinct while low-density populations survive (*Figure 2*, bottom left). For example, at a resistant fraction of 1/4, low-density populations will approach the survival fixed point while high-density populations will approach extinction as the separatrix is crossed. These counterintuitive dynamics, which we refer to as

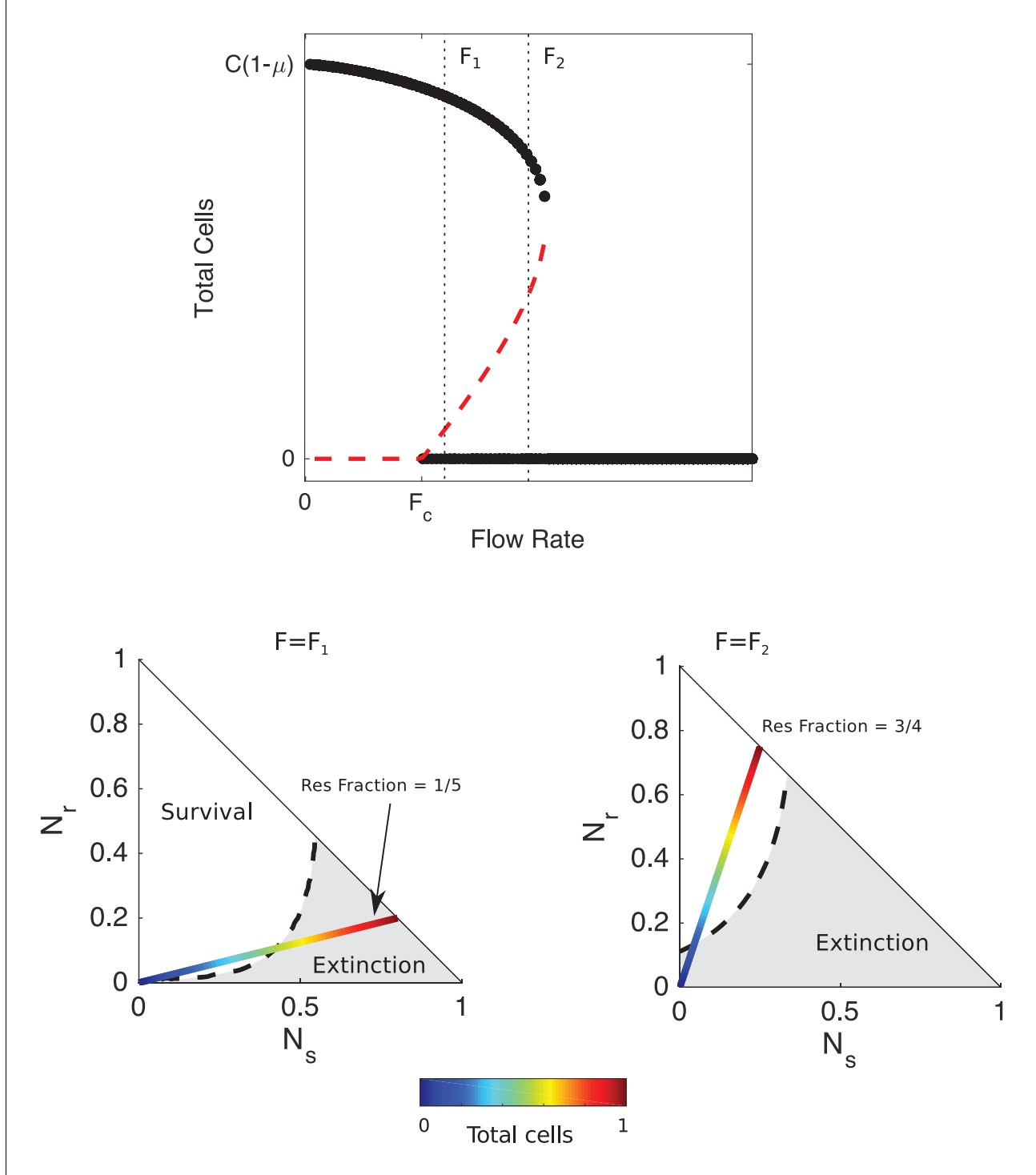

**Figure 2.** Mathematical model predicts bistability due to opposing density-dependent effects of sensitive and resistant cells on drug inhibition. Top: bifurcation diagram showing stable (filled circles) and unstable (red dashed curves) fixed points for different values of drug influx ($F$) and the total number of cells ($N_s + N_r$). $F_c \approx \mu K$ is the critical value of drug influx above which the zero solution (extinction) becomes stable; $\mu$ is the rate at which cells and drugs are removed from the system (and is measure in units of $g_{max}$, the maximum per capita growth rate of cells in drug-free media, and $K_r$ is the factor increase in drug MIC of the resistant strain relative to the wild-type strain. Vertical black dashed lines correspond to $F = F_1 > F_c$ (small drug influx, just above threshold) and $F_2 \gg F_c$ (large drug influx). Bottom panels: regions of survival (white) and extinction (grey) in the space of sensitive ($N_s$) and resistant ($N_r$) cells for flow rate $F = F_1$ (left) and $F = F_2$ (right). Dashed lines show separatrix, the contour separating survival from extinction. Multicolor lines represent constant resistant fractions (1/5, left; 3/4, right) at different total population sizes (ranging from 0 (blue) to a maximum density of 1 (red)).

*Figure 2 continued on next page*

Figure 2 continued

Cell numbers are measured in units of carrying capacity. Specific numerical plots were calculated with $h = 1.4$, $g_{min} = 1/3$, $g_{max} = 1$, $\epsilon_1 = 1.1$, $\epsilon_2 = 1.5$, $\gamma = 0.1$, $K_r = 14$, $F_1 = 1.4$, and $F_2 = 2.2$.

The online version of this article includes the following figure supplement(s) for figure 2:

**Figure supplement 1.** Separatrix contours separating survival from extinction with and without reverse inoculum effect.

'inverted bistability', are governed in part by the reverse inoculum effect, which leads to a rapid increase in drug efficacy in the high-density populations and a corresponding population collapse. Mathematically, the different behavior corresponds to a translation in the separatrix curve as the influx rate is modulated (*Figure 2*; see also *Figure 2—figure supplement 1*). Interestingly, the stable solutions that correspond to survival are comprised of only resistant cells. Hence, the model is not predicting a stable coexistence of sensitive and resistant strains (though such coexistence can exist under some conditions; *Lenski and Hattingh, 1986*); instead, the initial presence of sensitive cells positions the population within the basin of attraction of states (like collapse) that would not be favored in their absence.

To further characterize the dynamics of the model, we numerically solved the coupled equations (*Equations 1, 3*) for different initial compositions (resistant cell fraction) and different drug influx rates. In each case, we considered both high-density (OD = 0.6) and low-density (OD = 0.1) populations. As suggested by the bifurcation analysis (*Figure 2*), the model exhibits bistability over a range of drug influx rates (*Figure 3A*). The qualitative behavior within this bistable region can vary significantly. For small resistant fractions and low drug influx, bistability favors survival of low-density populations, while large resistant fractions and high drug influx favor survival of high-density populations. The parameter space is divided into four non-overlapping regions, leading to a phase diagram that predicts regions of extinction, survival, and bistabilities. These qualitative features are not unique the specific model described here, but also occur in alternative models that include, for example, more realistic Monod-style growth (SI; *Figure 3—figure supplement 1* through *Figure 3—figure supplement 2*). It is notable that the dynamics leading to the fixed points can be significantly more complex than simple mononotic increases or decreases in population size (*Figure 3A*, top panels).

## Small *E. faecalis* populations survive and large populations collapse when drug influx is slightly supercritical and resistant subpopulations are small

To test these predictions experimentally, we first performed a preliminary scan of parameter space in short, 5-hr experiments starting from a wide range of initial population fractions and drug influx rates (*Figure 3—figure supplement 3*). Based on these experiments, we then narrowed our focus to a region of 'high' influx rate ($F \approx 600 - 700 \; \mu g/mL/hr$), where conditions may favor 'normal' bistability, and a region of 'low' influx rate ($F \approx 15 - 20 \; \mu g/mL$ per hour), where conditions may favor 'inverted' bistability. Then, we performed replicate ($N = 3$) 20 hr experiments starting from a range of population compositions. Note that in the absence of density-mediated changes in drug concentration, these flow rates are expected to produce drug concentrations that increase over time, rapidly eclipsing the low-density limits for IC$_{50}$'s of both susceptible and resistant cells (see *Figure 1*) and exponentially approaching steady state values of $D = F/\mu \approx 8.5F$ with a time constant of $\mu^{-1} \approx 8.5$ (and therefore $\mu_0 = 8.5$ hr).

The experiments confirm the existence of both predicted bistable regimes as well as the expected regimes of survival and extinction (*Figure 3B–C*). At each of the two flow rates ($F_1$ and $F_2$), we observe a transition from density-independent extinction–where populations starting from both high and low-densities collapse–to density-independent survival–where both populations survive–as the initial resistant population is increased (*Figure 3B–C*, left to right). However, in both cases, there are intermediate regimes where initial population density determines whether the population will survive or collapse. When drug influx is relatively high ($F_2$) and the population is primarily comprised of resistant cells (55 percent), initially large populations survive while small populations collapse (*Figure 3B*, middle panel). On the other hand, when initial populations contain 11–15% resistant cells and drug influx is relatively small ($F_1$), we observe a clear region of 'inverted' bistability (*Figure 3C*, middle panel). In this regime, high-density populations (red) grow initially before undergoing

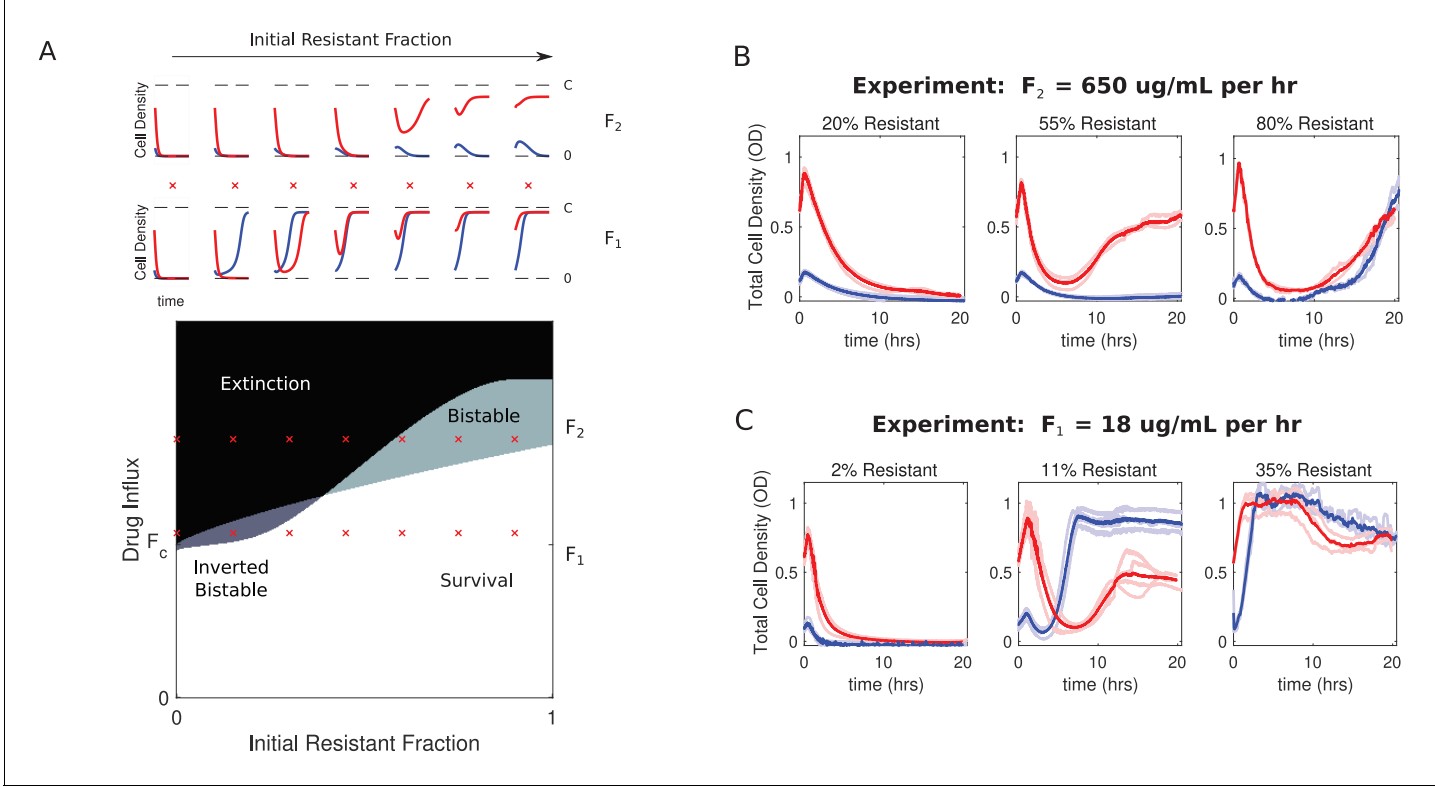

**Figure 3.** Bistability may favor survival of populations with highest or lowest initial density. (**A**) Main panel: phase diagram indicating regions of extinction (black), survival (white), bistability (light gray; initially large population survives, small population dies), and 'inverted' bistability (dark gray; initially small population survives, large population dies). Red 'x' marks correspond to the subplots in the top panels. Top panels: time-dependent population sizes starting from a small population (OD = 0.1, blue) and large population (OD = 0.6, red) at constant drug influx of $F_2 \gg F_c$ (large drug influx) and $F_1 > F_c$ (small drug influx). $F_c$ is the critical influx rate above which the extinct solution (population size 0) first becomes stable; it depends on model parameters, including media refresh rate ($\mu$), maximum kill rate of the antibiotic ($g_{min}$), the Hill coefficient of the dose response curve ($h$), and the MIC of the drug-resistant population in the low-density limit where cooperation is negligible ($K$). Specific numerical plots were calculated with $h = 1.4$, $g_{min} = 1/3$, $g_{max} = 1$, $\epsilon_1 = 1.1$, $\epsilon_2 = 1.5$, $\gamma = 0.1$, $K_r = 14$, $F_1 = 1.4$, and $F_2 = 2.2$. (**B**) Experimental time series for mixed populations starting at a total density of OD = 0.1 (blue) or OD = 0.6 (red). The initial populations are comprised of resistant cells at a total population fraction of 0.2 (left), 0.55 (center), and 0.80 right) for influx rate $F_1 = 650 \,\mu g/mL$. Light curves are individual experiments, dark curves are means across all experiments. (**C**) Experimental time series for mixed populations starting at a total density of OD = 0.1 (blue) or OD = 0.6 (red). The initial populations are comprised of resistant cells at a total population fraction of 0.02 (left), 0.11 (center), and 0.35 right) for influx rate $F_1 = 18 \,\mu g/mL$. Light curves are individual experiments, dark curves are means across all experiments.

The online version of this article includes the following source data and figure supplement(s) for figure 3:

**Source data 1.** Experimental data B in *Figure 3*.
**Source data 2.** Experimental data C in *Figure 3*.
**Figure supplement 1.** Alternative mathematical models exhibit similar qualitative features, including inverted bistability.
**Figure supplement 2.** Time series of cell density for simulations starting from high- or low-density populations in Monod growth model.
**Figure supplement 3.** Short experiments to explore parameter space for inverted bistability.
**Figure supplement 4.** Time series of cell density for simulations starting from high- or low-density populations in enzyme release model.
**Figure supplement 5.** Time series of cell density for simulations starting from high- or low-density populations in pH-IC$_{50}$ model.
**Figure supplement 6.** Time series of cell density for simulations starting from high or low-density populations.
**Figure supplement 7.** Isolates from populations exhibiting collapse but not complete extinction show similar dose-response behavior as original strains.
**Figure supplement 8.** Populations exhibit transient periods of approximately constant density near regions of inverted bistability.

dramatic collapse, while low-density populations (blue) initially decay before recovering and eventually plateauing near the carrying capacity. In contrast to predictions of the model, the collapsing populations do not entirely go extinct. We confirmed that these populations do indeed contain living cells, and single colony isolates exhibit dose-response characteristics similar to those of the original sensitive and resistant strains, so there is no evidence that additional resistance has evolved

during the experiment (*Figure 3—figure supplement 4*). Mathematical models do indicate the existence of long-lived but transient states of non-zero density near the onset of inverted bistability (*Figure 3—figure supplement 4*), which may partially explain the lack of complete extinction. However, it is also possible that it reflects features not included in the model. For example, while ampicillin is generally considered to be stable in solution for several days, the degradation rate depends on both temperature and pH (*Hou and Poole, 1969*), which could induce new dynamics on timescales of 10 s of hours. Similarly, β-lactamase activity can also vary slightly with pH, adding an additional layer of coupling between the density effects driven by sensitive and resistant cells (*Ohsuka et al., 1995*).

## Inverted bistability depends on pH-dependent reverse inoculum effect

The model predicts that the inverted bistability relies on the reverse inoculum effect–specifically, it requires $\varepsilon_1 > 0$ and is eliminated when $\varepsilon_1 = 0$ (*Figure 4*). Previous work showed that in this system, the reverse inoculum effect is driven by density-modulated changes in the local pH (*Karslake et al., 2016*). Conveniently, then, it is possible–in principle–to eliminate the effect by strengthening the buffering capacity of the media. To test this prediction, we repeated the experiments in the inverted bistable region in strongly buffered media (*Figure 4*). As predicted by the model, we no longer observe collapse of high-density populations, indicating that the region of inverted bistability is now a region of density-independent survival.

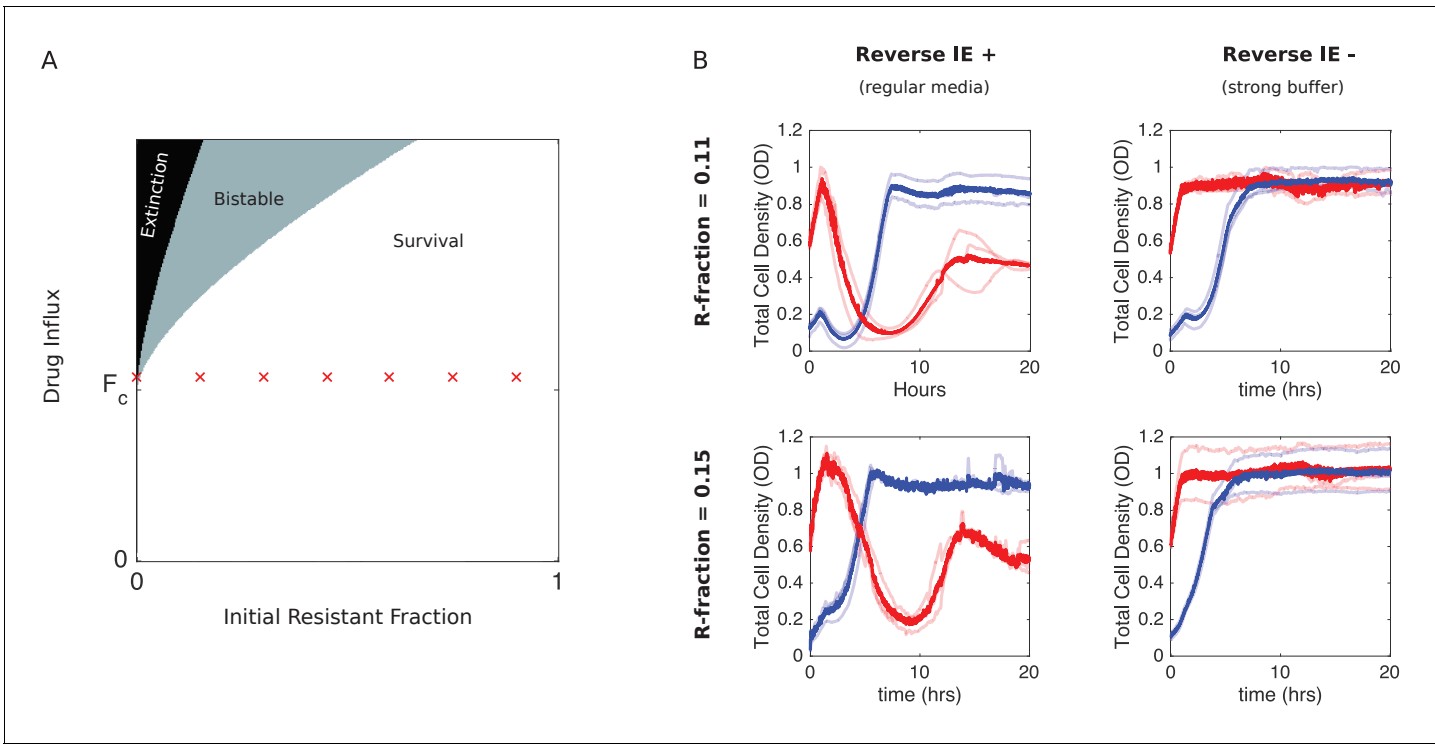

**Figure 4.** Eliminating reverse inoculum effect eliminates inverted bistability. (**A**) Numerical phase diagram in absence of reverse inoculum effect ($\epsilon_1 = 0$) indicating regions of extinction (black), survival (white), and bistability (light gray; initially large population survives, small population dies). There are no regions of 'inverted' bistability (initially small population survives, large population dies). Red 'x' marks fall along a line that previously traversed a region of inverted bistability in the presence of a reverse inoculum effect (*Figure 3*) but includes only surviving populations in its absence. $F_c$ is the critical influx rate above which the extinct solution (population size 0) first becomes stable; it depends on model parameters, including media refresh rate (μ), maximum kill rate of the antibiotic ($g_{min}$), the Hill coefficient of the dose-response curve (*h*), and the MIC of the drug-resistant population in the low-density limit where cooperation is negligible (*K*). Specific phase diagram was calculated with same parameters as in *Figure 3* except $\epsilon_1$, which corresponds to the reverse inoculum effect, is set to 0. (**B**) Experimental time series for mixed populations starting at a total density of OD = 0.1 (blue) or OD = 0.6 (red) in regular media (left panels) or strongly buffered media (right panels). The initial populations are comprised of resistant cells at a total population fraction of 0.11 (top) and 0.15 (bottom) and for influx rate of $F_1 = 18$ μg/mL. Light curves are individual experiments, dark curves are means across all experiments.

The online version of this article includes the following source data for figure 4:

**Source data 1.** Experimental data in *Figure 4*.

## Delaying antibiotic exposure can promote population collapse

The competing density-dependent effects on drug efficacy raise the question of whether different time-dependent drug dosing strategies might be favorable for populations with different starting compositions. In particular, we wanted to investigate the effect of delaying the start of antibiotic influx for different population compositions and influx rates. Based on the results of the model, we hypothesized that there would be two possible regimes where delay could dramatically impact survival dynamics: one (corresponding to 'normal bistability') where delaying treatment would lead to larger end-point populations, and a second (corresponding to"inverted bistabillity') where delaying treatment could, counterintuitively, promote population collapse (see *Figure 5—figure supplement 1*).

To test this hypothesis, we measured the population dynamics in mixed populations starting from an initial OD of 0.1 at time zero. We then compared final population size for identical populations experiencing immediate or delayed drug influx, with delay ranging from 0.5 to 2.5 hr. In experiments with non-zero delays, antibiotic influx was replaced by influx of drug-free media (at the same flow rate) during the delay period. In the first case, we chose a relatively small initial resistant fraction (0.11) and a relatively slow drug influx rate ($F = 18$ µg/mL), while in the second case we chose a larger initial resistant fraction (0.55) and a faster drug influx ($F = 650$ µg/mL).

Remarkably, we found that delaying treatment can have opposing effects in the two scenarios (*Figure 5*). At high drug influx rates and largely resistant populations, immediate treatment leads to smallest final populations (*Figure 5*, right panels), consistent with model predictions of bistability. Intuitively, the delay allows the subpopulation of resistant cells to increase in size, eventually surpassing a critical density where the presence of enzyme is sufficient to counter the inhibitory effects of antibiotic. On the other hand, at lower influx rates and lower initial resistant fractions, we find that immediate treatment leads to initial inhibition followed by a phase of rapid growth as the population thrives; by contrast, delays in treatment allow the population to initially grow rapidly before collapsing (*Figure 5*, left panels). It is particularly striking that delayed treatments–which also use significantly less total drug–can promote population collapse when immediate treatments appear to fail. Similar to the 'inverted bistability' observed earlier, the beneficial effects of delayed treatment can be traced to density-dependent drug efficacy–in words, the delay means the drug is applied when the population is sufficiently large that pH-mediated drug potentiation promotes collapse.

## Discussion

We have shown that different types of coupling between cell density and drug efficacy can lead to surprising dynamics in *E. faecalis* populations exposed to time-dependent ampicillin concentrations. In regimes of relatively fast or slow rates of drug influx, the results are intuitive: populations either survive or collapse, independent of initial population size (density). The intermediate regime, however, is characterized by bistability, meaning that population collapse will depend on initial population size. In regimes where cooperative resistance–in this case, due to enzymatic degradation of drug–dominates, larger populations are favored, similar to results predicted from the classical inoculum effect (*Udekwu et al., 2009*; *Karslake et al., 2016*). Under those conditions, it is critical to immediately expose cells to drug influx, as delays lead to increasingly resilient populations. Even more surprisingly, regimes characterized by comparatively smaller resistant populations and slower drug influx can lead to 'inverted bistability' where initially small populations thrive while large populations collapse. In this case, delays to drug exposure can paradoxically promote population collapse. It is notable that the mathematical model suggests these results are not simply transient effects but instead reflect asymptotic behavior where the system approaches one of two stable fixed points (survival or extinction) with very different biological consequences.

Our goal was to understand population dynamics in simple, single-species populations where environmental conditions–including drug influx rate and population composition–can be well controlled. To make sense of experimental results and, more importantly, to generate new testable hypotheses, we developed a minimal mathematical model and analyzed its qualitative behavior using standard tools from dynamical systems and bifurcation theory. We chose to focus on a phenomenological model in an effort to simplify the assumptions and limit the number of unconstrained parameters. However, our model clearly omits a number of potentially relevant biological details. For example, the model neglects evolutionary changes, such as de novo mutations, that would impact behavior on longer time-scales. Similarly, previous work (*Meredith et al., 2018*) has shown

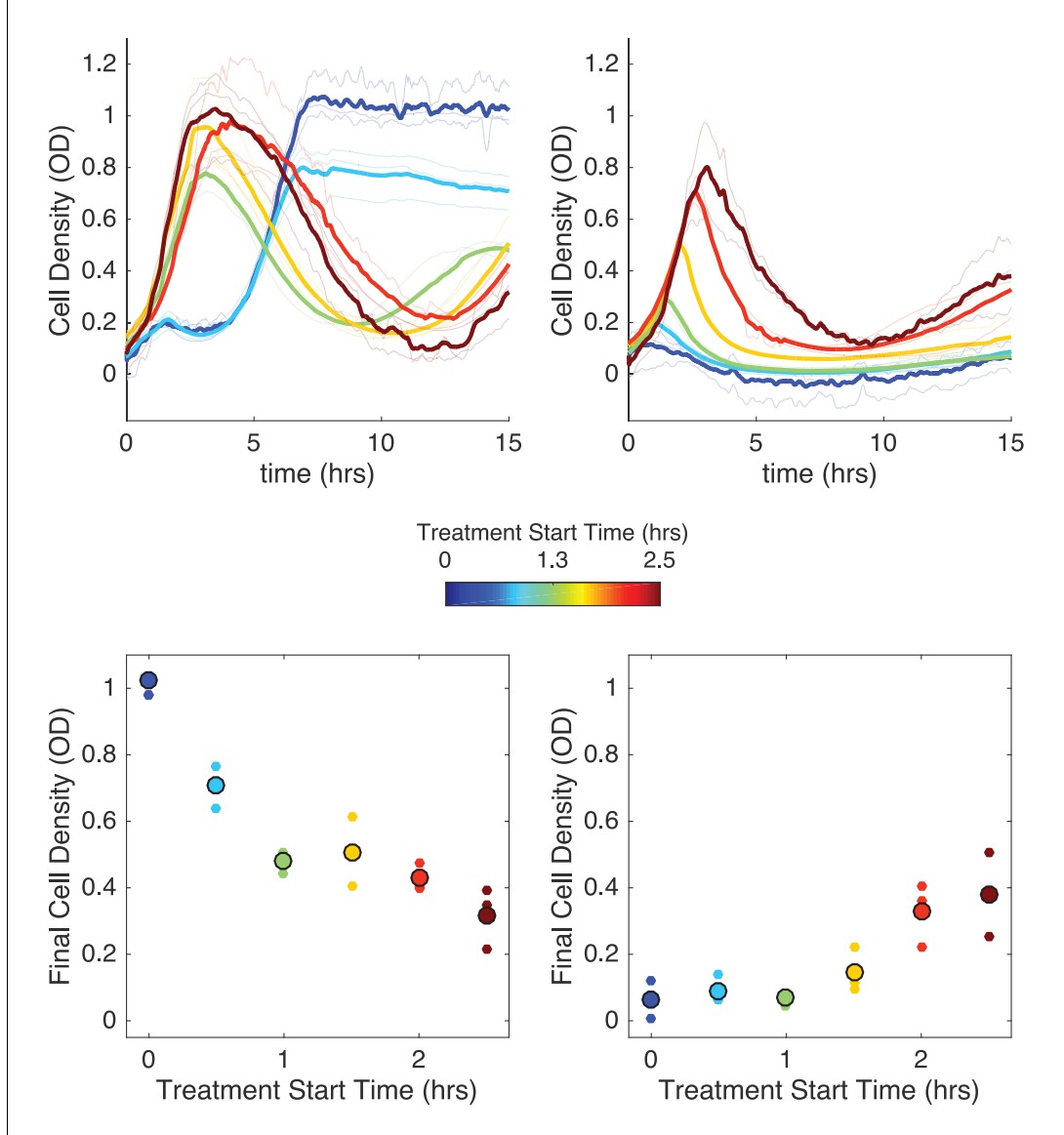

**Figure 5.** Delaying antibiotic exposure tips populations toward survival or extinction depending on initial resistance fraction and drug influx rate. Top panels: experimental time series for mixed populations with small initial resistance and low drug influx (left; initial resistance fraction, 0.11 μg/mL) or large initial resistance and high drug influx (right; initial resistance fraction, 0.55 μg/mL). Antibiotic influx was started immediately (blue) or following a delay of up to 2.5 hr (dark red). Light transparent lines are individual replicates; dark lines are means over replicates. In experiments with nonzero delays, antibiotic influx was replaced by influx of drug-free media during the delay. Bottom panels: final cell density (15 hr) as a function of delay ('treatment start time'). small points are individual replicates; large circles are means across replicates.

The online version of this article includes the following source data and figure supplement(s) for figure 5:

**Source data 1.** Experimental data in *Figure 5*.

**Figure supplement 1.** Numerical results indicate that delaying antibiotic exposure tips populations toward survival or extinction depending on initial resistance fraction and drug influx rate.

that lysis of resistant cells can effectively increase the concentration of drug-degrading enzyme. We find that extending our phenomenological model to account for free enzyme leads to qualitatively similar behavior (see SI), but more accurate kinetic models may point to different dynamics in some regimes. Constructing detailed mechanistic models is notoriously difficult, but recent work shows that careful pairing of experiment and theory can be used to systematically overcome many common obstacles (*Hart et al., 2019*). A similar approach could potentially be applied to this system, leading to more accurate quantitative models that account for factors like spontaneous drug degradation (*Hou and Poole, 1969*), the pH dependence of β-lactamase activity (*Ohsuka et al., 1995*), and the kinetics of pH-modulated drug activity.

It is obvious that the specific in vitro conditions used here fail to capture numerous complexities associated with resistance in clinical settings (*Bonten et al., 2001*), including substantial spatial heterogeneity, potential for biofilm formation, effects of the host immune system, and drug concentrations that differ in both magnitude and time-course from the specific scenarios considered here. In particular, the effects of delayed antibiotic exposure in a clinical setting will depend on many factors not captured here, and there are unquestionably scenarios where such delay could be detrimental to patient well-being. In fact, our results indicate that delaying drug exposure can have differing effects in different parameter regimes, even in laboratory populations. Future work in clinically motivated in vitro systems, such as biofilms, and ultimately in vivo are needed to assess the feasibility of delayed dosing in more realistic scenarios. In addition, we note that β-lactamase producing enterococci are thought to be relatively rare, though they have been associated with multi-drug resistant, high-risk enterococcal infections (*Murray, 1992*; *Wells et al., 1992*; *Arias et al., 2010*) and may be more widespread that initially believed because of the difficulty of detection in traditional laboratory tests (*Gagetti et al., 2019*). Finally, our experimental model system is based on plasmid-mediated resistance, and while this fact is not explicitly assumed in any of our mathematical models, horizontal gene transfer may introduce new dynamics (*Lopatkin et al., 2016*; *Lopatkin et al., 2017*), particularly in high-density populations where conjugation is frequent.

Our results show that the response of microbial populations to antibiotic can be surprisingly complex, suggesting that the spread of resistance alleles may not always follow simple selection dynamics. These findings underscore the need for additional metrics (similar to the proposed notion of drug resilience; *Meredith et al., 2018*) that go beyond short-term growth measurements to for population dynamics over multiple timescales. More generally, we hope these results will motivate continued efforts to understand the potentially surprising ways that molecular level resistance events influence dynamics on the scale of microbial populations.

# Materials and methods

## Key resources table

| Reagent (species) | Designation | Source | Additional info |
|---|---|---|---|
| Gene (*E. faecalis*) | β-lactamase | *Zscheck and Murray, 1991*; *Rice et al. (1991)*; *Rice and Marshall (1992)* | PCR from strain CH19 |
| Gtrain (*E. faecalis*) | OG1RF | *Dunny et al. (1978)*; *Oliver et al. (1977)* | |
| Plasmid | pBSU101-DasherGFP | *Aymanns et al. (2011)*; *Hallinen et al. (2019)* | Reporter plasmid |
| Plasmid | pBSU101-BFP-BL | *Hallinen et al. (2019)*, this paper | Expresses β-lactamase |
| Drug | Spectinomcyin sulfate | MP Biomedicals | CAT 0215899302 |
| Drug | Ampicillin Sodium Salt | Fisher | CAT BP1760-25 |

## Bacterial strains, media, and growth conditions

Experiments were performed with *E. faecalis* strain OG1RF, a fully sequenced oral isolate. Ampicillin-resistant strains were engineered by transforming (*Dunny et al., 1991*) OG1RF with a modified version of the multicopy plasmid pBSU101, which was originally developed as a fluorescent reporter for Gram-positive bacteria (*Aymanns et al., 2011*). The plasmid was chosen because it can be conveniently manipulated and propagated in multiple species (including *E. coli*) and contains a fluorescent reporter that provides a redundant control for readily identifying the strains. The modified

plasmid, named pBSU101-BFP-BL, expresses BFP (rather than GFP in the original plasmid) and also constitutively expresses -lactamase driven by a native promoter isolated from the chromosome of clinical strain CH19 (*Rice et al., 1991*; *Rice and Marshall, 1992*). The β-lactamase gene and reporter are similar to those found in other isolates of enterococci and streptococci (*Murray and Mederski-Samaroj, 1983*; *Zscheck and Murray, 1991*). Similarly, sensitive strains were transformed with a similar plasmid, pBSU101-DasherGFP, a pBSU101 derivative that lacks the β-lactamase insert and where eGFP is replaced by a brighter synthetic GFP (Dasher-GFP; ATUM ProteinPaintbox, https://www.atum.bio/). The plasmids also express a spectinomycin resistance gene, and all media was therefore supplemented with spectinomycin.

## Antibiotics

Antibiotics used in this study included Spectinomycin Sulfate (MP Biomedicals) and Ampicillin Sodium Salt (Fisher).

## Estimating IC$_{50}$ for sensitive and resistant strains

Experiments to estimate the half-maximal inhibitory concentration (IC$_{50}$) for each population were performed in 96-well plates using an Enspire Multimodal Plate Reader. Overnight cultures were diluted $10_2$ - $10_8$ fold into individual wells containing fresh BHI and a gradient of 6–14 drug concentrations. After 20 hr of growth, the optical density at 600 nm (OD) was measured and used to create a dose response curve, which was fit to a Hill-like function $f(x) = (1 + (x/K)^h)^{-1}$ using nonlinear least squares fitting, where $K$ is the half-maximal inhibitory concentration (IC$_{50}$) and $h$ is a Hill coefficient describing the steepness of the dose-response relationship.

## Continuous culture device

Experiments were performed in custom-built, computer-controlled continuous culture devices (CCD) as described in *Karslake et al. (2016)*. Briefly, bacterial populations are grown in glass vials containing a fixed volume of 17 mL media. Cell density was measured at 1.5 s intervals in each vial using emitter/detector pairs of infrared LEDs (Radioshack). Detectors register a voltage output that is then converted to optical density using a calibration curve performed with a table top OD reader. Each vial contains input and output channels connected to silicone tubing and attached to a system of peristaltic pumps (Boxer 15000, Clark Solutions) that add drug and/or media and remove excess liquid on a schedule that can be programmed in advance or determined in real time. The entire system is controlled using a collection of DAQ and instrument control modules (Measurement Computing) along with the Matlab (MathWorks) Instrument Control Toolbox.

## Drug dosing protocols

In 'constant flow' experiments, media (with drug, when relevant) is added at a rate of 1 mL/min for a total of 7.5 s every 3.75 min for an effective flow rate of 2 mL/hr (corresponding to a rate constant of μ = 10.12 hr$^{-1}$ in 17 mL total volume). Media (plus cells and drug) is removed at an identical rate to maintain constant volume. While drug influx (and waste removal) strictly occurs on discrete on-off intervals, the timescale of those intervals (3–4 min) is an order of magnitude slower than the maximum bacterial growth rate under these conditions, which corresponds to a doubling time of approximately 30–40 min. The influx of drug is therefore approximately continuous on the timescale of bacterial dynamics. We experimentally modulate the influx rate of drug, $F$, without changing the background refresh rate (μ) by changing the drug concentration in the drug reservoir. For experiments involving time-dependent drug influx–for example, those in *Figure 5*, the media in the drug reservoirs is exchanged manually at specified times to mimic, for example, delayed treatment start times.

## Experimental mixtures and set up

All experiments were started from overnight cultures inoculated from single colonies grown on BHI agar plates with streptomycin and incubated in sterile BHI (Remel) with streptomycin (120 μg/mL) overnight at 37C. Highly buffered media was prepared by supplementing standard BHI with 50 μM Dibasic Sodium Phosphate (Fisher). Overnight cultures were diluted 100–200 fold with fresh BHI in continuous culture devices and populations were allowed to reach steady state exponential growth

at the specified density (typically OD = 0.1 or OD = 0.6) prior to starting influx and outflow of media and waste. Experiments were typically performed in triplicate.

## Acknowledgements

This work is supported by the National Science Foundation (NSF No. 1553028 to KW; NSF GRF to KH), the National Institutes of Health (NIH No. 1R35GM124875-01 to KW), and the Hartwell Foundation for Biomedical Research (to KW).

## Additional information

### Funding

| Funder | Grant reference number | Author |
|---|---|---|
| National Science Foundation | 1553028 | Kevin B Wood |
| National Institute of General Medical Sciences | 1R35GM124875 | Kevin B Wood |
| National Science Foundation | GRFP | Kelsey M Hallinen |

The funders had no role in study design, data collection and interpretation, or the decision to submit the work for publication.

### Author contributions

Kelsey M Hallinen, Jason Karslake, Conceptualization, Formal analysis, Investigation, Visualization; Kevin B Wood, Conceptualization, Formal analysis, Supervision, Funding acquisition, Visualization

### Author ORCIDs

Kelsey M Hallinen (iD) http://orcid.org/0000-0003-4081-6699
Kevin B Wood (iD) https://orcid.org/0000-0002-0985-7401

### Decision letter and Author response

Decision letter https://doi.org/10.7554/eLife.52813.sa1
Author response https://doi.org/10.7554/eLife.52813.sa2

## Additional files

### Supplementary files

• Transparent reporting form

### Data availability

All experimental data are included in the manuscript and supporting files.

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

## Appendix 1

### Alternative Models

In what follows, we outline three alternative but related models, each of which exhibits similar qualitative behavior to the one proposed in the main text (*Figure 3—figure supplement 1*). More specifically, all models are characterized by a region of extinction for sufficiently large drug influx, survival for sufficiently low drug influx, and a region of bistable behavior for intermediate rates of influx. The bistable regime in all cases consists of a region of 'inverted bistability', where low density populations survive while high-density populations collapse, and a region of 'normal bistability' where low density populations collapse while high-density populations survive. In all cases, the inverted bistable region occurs in regions of smaller drug influx and lower initial resistant fraction, while normal bistability occurs for larger rates of influx and higher initial resistant fractions.

### Enzyme Release Model

Recent work has shown that lysis of enzyme-producing cells may lead to the release of enzyme into the media, which can continue to degrade drug even in the absence of the original (now lysed) cell. To capture this effect, we modify the original model to

$$
\begin{aligned}
\frac{dN_s}{dt} &= g(D)\left(1 - \tfrac{N_s + N_r}{C}\right)N_s - \mu N_s, \\
\frac{dN_r}{dt} &= g(D')\left(1 - \tfrac{N_s + N_r}{C}\right)N_r - \mu N_r, \\
\frac{dD}{dt} &= F + \epsilon_1(N_s + N_r)D - \epsilon_2 N_r D - D\mu - \chi E D, \\
\frac{dE}{dt} &= r_{death}\left(1 - \tfrac{N_s + N_r}{C}\right)N_r - E\mu,
\end{aligned}
\tag{4}
$$

where $E$ describes lysed (resistant) cells, $r_{death}$ is the death rate of resistant cells due to antibiotic ($r_{death} \equiv g_{max} - g(D')$) and $\chi$ is a parameter that sets the degradation rate of antibiotic per lysed cell (due to cell-free enzyme). We take $\chi = 0.1$ for the simulations in *Figure 3—figure supplement 1*; *Figure 3—figure supplement 4*.

As before, $N_s$ is the density of sensitive cells, $N_r$ the density of resistant cells, $C$ is the carrying capacity (set to one without loss of generality), μ is a rate constant that describes the removal of cells due to (slow) renewal of media and addition of drug, $D$ is the effective concentration of drug (measured in units of MIC of the sensitive cells), and $D' = D/K_r$, where $K_r$ is a factor that describes the increase in drug minimum inhibitory concentration (MIC) for the resistant (enzyme producing) cells in low density populations where cooperation is negligible, and the function $g(x)$ is a dose response function with parameters $h$ (a Hill coefficient), $g_{max}$ (the growth in the absence of drug), and $g_{min} > 0$ (the maximum death rate). In addition, $\epsilon_1 > 0$ is an effective rate constant describing the reverse inoculum effect (proportional to total population size), $\epsilon_2 > 0$ describes the enzyme-driven 'normal' inoculum effect, and $F = D_r\mu$ is rate of drug influx into the reservoir, which can be adjusted by changing the concentration $D_r$ in the drug reservoir.

### pH-IC$_{50}$ Model

The original model captures the pH-driven rIE using a density-dependent increase in the effective drug concentration. An alternative, and perhaps more intuitive (but related) phenomenological model assumes that the rIE leads directly to a density dependent change in the IC$_{50}$ of the drug. In principle, the dependence of IC$_{50}$ on density can be estimated from independent experiments, leading to a more quantitatively constrained model (similar to the approach in *Hart et al., 2019*). Specifically, we have

$$\frac{dN_s}{dt} = g(D/K_d(n))\left(1 - \frac{N_s+N_r}{C}\right)N_s - \mu N_s,$$

$$\frac{dN_r}{dt} = g(D'/K_d(n))\left(1 - \frac{N_s+N_r}{C}\right)N_r - \mu N_r, \tag{5}$$

$$\frac{dD}{dt} = F - \epsilon_2 N_r D - D\mu,$$

where we have now explicitly noted the dependence of the IC$_{50}$, which we denote $K_d(n)$, on the total cell density $n = N_s + N_r$. For simplicity, we take

$$K_d(n) = \frac{K_0}{1 + n/\epsilon_3} \tag{6}$$

where $K_0$ is the low density limit of the IC$_{50}$ and $\epsilon_3$ sets the density scale at which the pH effects are half maximal. We take $K_0 = 1$ and $\epsilon_3 = 1/5$ for the simulations in **Figure 3—figure supplement 1**; **Figure 3—figure supplement 5**.

## Monod Growth Model

Because experiments take place in a chemostat, a more realistic description of growth is given by the classic Monod growth model, where growth is explicitly exponential (rather than logistic) but the rate depends on the concentration of a limiting nutrient $S$ (**Edelstein-Keshet, 2005**; **Allen and Waclaw, 2019**). Specifically, we have

$$\frac{dN_s}{dt} = g(D)g_M(S)N_s - \mu N_s,$$

$$\frac{dN_r}{dt} = g(D')g_M(S)N_r - \mu N_r,$$

$$\frac{dD}{dt} = F + \epsilon_1(N_s + N_r)D - \epsilon_2 N_r D - D\mu, \tag{7}$$

$$\frac{dS}{dt} = \mu(C_0 - S) - \eta(N_s + N_r)g_M(S),$$

where we have introduced a new equation for $S$, the limiting nutrient $S$, and where $C_0$ is the concentration of nutrient in the stock (influx) reservoir and $g_M(S)$ is the nutrient-dependent Monod growth, given by

$$g_M(S) = \frac{S}{S + K_s}. \tag{8}$$

For simplicity, we take $C_0 = K_s = \eta = 1$, which leads to a drug-free steady state cell density similar to that in the other models (**Figure 3—figure supplement 1**; **Figure 3—figure supplement 2**).

