## [Decision Letter]

**Acceptance summary:**

Hallinen et al. combine modeling and experiments to show that the density-dependent effects of antibiotic degradation by resistant cells and the pH influence on antibiotic survival can result in complex and counter-intuitive dynamics.

**Decision letter after peer review:**

Thank you for submitting your article "Delayed antibiotic exposure induces population collapse in enterococcal communities with drug-resistant subpopulations" for consideration by *eLife*. Your article has been reviewed by three peer reviewers, including Wenying Shou as the Reviewing Editor and Reviewer #1, and the evaluation has been overseen by Aleksandra Walczak as the Senior Editor. The following individuals involved in review of your submission have agreed to reveal their identity: Lingchong You (Reviewer #2).

The reviewers have discussed the reviews with one another and the Reviewing Editor has drafted this decision to help you prepare a revised submission.

Summary:

All three reviewers are positive about your work, although we feel that stating the limitations of your study is important. We are also confused about Figure 4C.

I attach modified full reviews to help with your revision.

Reviewer #1:

Hallinen et al. showed that the density-dependent effects of antibiotic degradation by resistant cells and the pH influence on antibiotic survival can result in complex and counter intuitive dynamics. Overall, I find the work quite interesting (although I am not intimately familiar with the field).

Major comments:

How mechanistic a model is a continuum. A phenomenological model was presented in this study. Have you ever tried constructing a more mechanistic model which would account for important aspects such as enzyme degradation of antibiotics and the pH effects? You can then use this model to explain reverse inoculum effect and possibly other experimental results. In the Discussion section, you delineated why a mechanistic model was not constructed, but I am not convinced, especially since you did not measure all parameters in your phenomenological model anyways. If you have tried to construct a more mechanistic model but it did not work, I think that it is still worth including such efforts in the Discussion section as well as speculations on why it might have failed. If you have not tried constructing a mechanistic model, you can cite examples where mechanistic modeling has been shown to be challenging (e.g. Hart et al., 2019).

Figure 4C middle panel: The red curve did not really go extinct, unlike the model prediction in Figure 3 or Figure 4A. Need to explain this.

Reviewer #2:

In this work, the authors investigate the population-level implications of two density-dependent effects that are known to influence bacterial resistance to β-lactam antibiotics (mediated by β-lactamase production). The first, the inoculum effect (IE), suggests that higher densities of resistant cells can decay more drug and therefore reduce its efficacy, while the second, the reverse inoculum effect (RIE), suggests that higher bacterial densities modulate the environment in ways that increase the sensitivity of populations to the drug. They probe this question using both mathematical modeling and bioreactor experiments with a mixed population of sensitive and resistant strains responding to ampicillin treatment. The authors observe both effects in their system and suggest that these competing effects can lead to complex dynamics, especially regions of bistability which implies that under different drug concentrations different initial densities will be favored. The authors attribute the reverse inoculum effect to pH effects, removing it with buffered media. Finally, they exploit this density dependence for populations where the inoculum effect is small (few resistant cells) by delaying treatment until the population density is high enough that the reverse inoculum effect dominates, and show that this delay can allow for population extinction where immediate treatment cannot.

I enjoyed reading the fascinating study, which underscores the rich population dynamics resulting from the interplay between bacterial populations and antibiotics. In particular, what's really interesting is how the composition of a mixture can dictate qualitatively opposite outcomes. The authors elegantly integrate analysis by a coarse-grained model and quantitative experiments throughout the study. The observed dynamics have implications for the effective design of antibiotic dosing when combating bacterial pathogens.

Reviewer #3:

The manuscript by Hallinen et al. is a well-crafted story involving an integration between computer-controlled bioreactors and simple mathematical models that reveals density-dependent feedback loops that address nonintuitive community-level behaviors upon antibiotic treatment. In particular it deals with β-lactam treatment of the pathogen *E. faecalis* when communities include a drug-resistant subpopulation that expresses a β-lactamase. The authors do a careful job of addressing the effects of treatment on resistant and sensitive populations as a function of density (either alone or in combination), and as a result are able to reveal some behaviors that highlight potentially important variables for driving resistant populations extinct.

A key aspect of their system is the reverse inoculum effect of β-lactams in which there is increased growth at lower densities, arising from changes in local pH. The authors do a good of breaking down the system into its component parts, demonstrating the rIE for their system and using this to motivate their mathematical model, which predicts a region of inverted bistability in which there is a big increase in drug efficacy in the high-density populations and a population collapse.

The work is well done and the paper is well written. I have a few comments/questions:

In Figure 4C (middle), the red population does undergo a large collapse, as stated in the paper in subsection “Small *E. faecalis* populations survive and large populations collapse when drug influx is slightly supercritical and resistant subpopulations are small”, but then it recovers. Why is this? Is there further resistance selection leading to other mutations (perhaps changes to the level of expression of the β-lactamase?)

One thing that pops out from their manuscript is that the conclusions are almost too clear once you know that there is both an inoculum effect and a reverse inoculum effect; nonetheless, it is good to see that the experiments can indeed achieve the predicted inverted bistability.

They take advantage of the inverted bistability to show that there are beneficial effects of delayed drug treatment. This is an interesting idea, but also very context dependent – you have to know a lot about drug concentration, relative populations of resistant/sensitive pools, initial density. It seems like this knowledge is important, but really hard to apply in a practical way. [This needs to be explicitly discussed in Discussion section.]

One thing the authors could do to address the point above is to do an experiment in a biofilm, where if they can see a similar inverted bistability, that would go a long way toward suggesting that this knowledge can be broadly useful. [Since this would be too substantial of a request, you could include this in the subsection “Future directions”.]

---

## [Author Response]

Summary:All three reviewers are positive about your work, although we feel that stating the limitations of your study is important. We are also confused about Figure 4C.I attach modified full reviews to help with your revision.

Thank you for the positive evaluations and helpful feedback. In particular, we believe the expanded Discussion section of the study’s limitations has improved manuscript.

Reviewer #1:Hallinen et al. showed that the density-dependent effects of antibiotic degradation by resistant cells and the pH influence on antibiotic survival can result in complex and counter intuitive dynamics. Overall, I find the work quite interesting (although I am not intimately familiar with the field).

We’re happy that the reviewer finds the work interesting.

Major comments:How mechanistic a model is a continuum. A phenomenological model was presented in this study. Have you ever tried constructing a more mechanistic model which would account for important aspects such as enzyme degradation of antibiotics and the pH effects? You can then use this model to explain reverse inoculum effect and possibly other experimental results. In the Discussion section, you delineated why a mechanistic model was not constructed, but I am not convinced, especially since you did not measure all parameters in your phenomenological model anyways. If you have tried to construct a more mechanistic model but it did not work, I think that it is still worth including such efforts in the Discussion section as well as speculations on why it might have failed. If you have not tried constructing a mechanistic model, you can cite examples where mechanistic modeling has been shown to be challenging (e.g. Hart et al., 2019).

The reviewer makes an excellent point, and it’s one we struggle with often. We ultimately decided to focus on a phenomenological model because many of the kinetic parameters needed for a more mechanistic model would be difficult to constrain, particularly since the density driven pH effect is not fully understood at the molecular level. Nevertheless, we recognize the limitations of phenomenological models, which often lack the specificity to make quantitatively accurate predictions. We have now added an expanded discussion of these points in the Discussion section, including a reference to the suggested paper where a combination of experiment and theory were used to overcome these obstacles. Specifically, we say:

“Our goal was to understand population dynamics in simple, single-species populations where environmental conditions–including drug influx rate and population composition–can be well controlled. To make sense of experimental results and, more importantly, to generate new testable hypotheses, we developed a minimal mathematical model and analyzed its qualitative behavior using standard tools from dynamical systems and bifurcation theory. We chose to focus on a phenomenological model in an effort to simplify the assumptions and limit the number of unconstrained parameters. However, our model clearly omits a number of potentially relevant biological details. For example, the model neglects evolutionary changes, such as de novo mutations, that would impact behavior on longer time-scales. Similarly, previous work (67) has shown that lysis of resistant cells can effectively increase the concentration of drug- degrading enzyme. We find that extending our phenomenological model to account for free enzyme leads to qualitatively similar behavior (see SI), but more accurate kinetic models may point to different dynamics in some regimes. Constructing detailed mechanistic models is notoriously difficult, but recent work shows that careful pairing of experiment and theory can be used to systematically overcome many common obstacles (68). A similar approach could potentially be applied to this system, leading to more accurate quantitative models that account for factors like spontaneous drug degradation (65), the pH dependence of β-lactamase activity (66), and the kinetics of pH-modulated drug activity.”

In addition, to show that the qualitative features of the model are not specific to formulation we chose, we added a new section to the SI that discusses several alternative models, including (1) a model that includes drug decay due to enzymes released upon lysis of resistant cells, (2) a model that incorporates the pH effect by directly making the drug’s IC_50_ dependent on cell density, and (3) a model that incorporates a more realistic Monod-like growth. In all 3 cases, the system exhibits qualitatively similar phase diagrams, though the quantitative features will of course vary between models and as parameters are changed (Figure 3—figure supplement 1, Figure 3—figure supplement 2, Figure 3—figure supplement 4, Figure 3—figure supplement 5).

Figure 4C middle panel: The red curve did not really go extinct, unlike the model prediction in Figure 3 or Figure 4A. Need to explain this.

Thank you for pointing out this discrepancy. We verified experimentally that these populations do indeed contain living cells (e.g. living colonies on plates). We also measured dose response curves for 6 isolates from these populations; 4 showed dose response behavior similar to that of the original sensitive strains, and 2 showed behavior similar to that of the original resistant strains (Figure 3—figure supplement 7). So we do not believe the lack of extinction is due to the emergence of new resistance mutations. A second possible explanation is that these small but nonzero populations represent a long-lived transient state— and indeed these states are observed in the model when parameters are near the “inverted” bistable regime (Figure 3—figure supplement 8). Unfortunately, for technical reasons it is difficult to run the experiment for longer than approximately 24 hours, so we are not able to distinguish experimentally between longlived transient states and stable steady states. Finally, it’s also possible that this behavior arises from biological or chemical dynamics that are not included in the model. We have now added the following discussion of this point in Results section:

“In contrast to predictions of the model, the collapsing populations do not entirely go extinct. We confirmed that these populations do indeed contain living cells, and single colony isolates exhibit dose-response characteristics similar to those of the original sensitive and resistant strains, so there is no evidence that additional resistance has evolved during the experiment (Figure X). Mathematical models do indicate the existence of long-lived but transient states of non-zero density near the onset of inverted bistability (Figure X), which may partially explain the lack of complete extinction. However, it is also possible that it reflects features not included in the model. For example, while ampicillin is generally considered to be stable in solution for several days, the degradation rate depends on both temperature and pH (65), which could induce new dynamics on timescales of 10s of hours. Similarly, β-lactamase activity can also vary slightly with pH, adding an additional layer of coupling between the density effects driven by sensitive and resistant cells (66).”

Reviewer #2:In this work, the authors investigate the population-level implications of two density-dependent effects that are known to influence bacterial resistance to β-lactam antibiotics (mediated by β-lactamase production). The first, the inoculum effect (IE), suggests that higher densities of resistant cells can decay more drug and therefore reduce its efficacy, while the second, the reverse inoculum effect (RIE), suggests that higher bacterial densities modulate the environment in ways that increase the sensitivity of populations to the drug. They probe this question using both mathematical modeling and bioreactor experiments with a mixed population of sensitive and resistant strains responding to ampicillin treatment. The authors observe both effects in their system and suggest that these competing effects can lead to complex dynamics, especially regions of bistability which implies that under different drug concentrations different initial densities will be favored. The authors attribute the reverse inoculum effect to pH effects, removing it with buffered media. Finally, they exploit this density dependence for populations where the inoculum effect is small (few resistant cells) by delaying treatment until the population density is high enough that the reverse inoculum effect dominates, and show that this delay can allow for population extinction where immediate treatment cannot.I enjoyed reading the fascinating study, which underscores the rich population dynamics resulting from the interplay between bacterial populations and antibiotics. In particular, what's really interesting is how the composition of a mixture can dictate qualitatively opposite outcomes. The authors elegantly integrate analysis by a coarse-grained model and quantitative experiments throughout the study. The observed dynamics have implications for the effective design of antibiotic dosing when combating bacterial pathogens.

We’re delighted that the reviewer finds the work fascinating and supports publication.

Reviewer #3:The manuscript by Hallinen et al. is a well-crafted story involving an integration between computer-controlled bioreactors and simple mathematical models that reveals density-dependent feedback loops that address nonintuitive community-level behaviors upon antibiotic treatment. In particular it deals with β-lactam treatment of the pathogen *E. faecalis* when communities include a drug-resistant subpopulation that expresses a β-lactamase. The authors do a careful job of addressing the effects of treatment on resistant and sensitive populations as a function of density (either alone or in combination), and as a result are able to reveal some behaviors that highlight potentially important variables for driving resistant populations extinct.A key aspect of their system is the reverse inoculum effect of β-lactams in which there is increased growth at lower densities, arising from changes in local pH. The authors do a good of breaking down the system into its component parts, demonstrating the rIE for their system and using this to motivate their mathematical model, which predicts a region of inverted bistability in which there is a big increase in drug efficacy in the high-density populations and a population collapse.The work is well done and the paper is well written. I have a few comments/questions:

We thank the reviewer for the positive evaluation.

In Figure 4C (middle), the red population does undergo a large collapse, as stated in the paper in subsection “Small *E. faecalis* populations survive and large populations collapse when drug influx is slightly supercritical and resistant subpopulations are small”, but then it recovers. Why is this? Is there further resistance selection leading to other mutations (perhaps changes to the level of expression of the β-lactamase?)

Please see our response to the major comments from reviewer 1 where we answer the same question.

One thing that pops out from their manuscript is that the conclusions are almost too clear once you know that there is both an inoculum effect and a reverse inoculum effect; nonetheless, it is good to see that the experiments can indeed achieve the predicted inverted bistability.

We are happy that the reviewer believes the conclusions are clear and appreciates the experimental verification.

They take advantage of the inverted bistability to show that there are beneficial effects of delayed drug treatment. This is an interesting idea, but also very context dependent – you have to know a lot about drug concentration, relative populations of resistant/sensitive pools, initial density. It seems like this knowledge is important, but really hard to apply in a practical way. [This needs to be explicitly discussed in Discussion section.]One thing the authors could do to address the point above is to do an experiment in a biofilm, where if they can see a similar inverted bistability, that would go a long way toward suggesting that this knowledge can be broadly useful. [Since this would be too substantial of a request, you could include this in the subsection “Future directions”.]

The two points above are well taken. We have added a substantial discussion of the limitations of the study, particularly in the context of delayed drug treatment in clinical settings. We have also included the study of biofilms in a discussion of future work. Specifically, we now write:

“It is obvious that the specific in vitro conditions used here fail to capture numerous complexities associated with resistance in clinical settings (69), including substantial spatial heterogeneity, potential for biofilm formation, effects of the host immune system, and drug concentrations that differ in both magnitude and time-course from the specific scenarios considered here. In particular, the effects of delayed antibiotic exposure in a clinical setting will depend on many factors not captured here, and there are unquestionably scenarios where such delay could be detrimental to patient well- being. In fact, our results indicate that delaying drug exposure can have differing effects in different parameter regimes, even in laboratory populations. Future work in clinically motivated in vitro systems, such as biofilms, and ultimately in vivo are needed to assess the feasibility of delayed dosing in more realistic scenarios.”